



# Specific Climate Classification for Mediterranean Hydrology and Future Evolution Under Med-CORDEX RCM Scenarios

Antoine Allam[1,2], Roger Moussa[2], Wajdi Najem[1], Claude Bocquillon[1]

[1]CREEN, Saint-Joseph University, Beirut, 1107 2050, Lebanon.

[2]LISAH, Univ. Montpellier, INRAE, IRD, SupAgro, Montpellier, France.

*Correspondence to* Antoine Allam (antoine_allam@hotmail.com)

**Abstract.** The Mediterranean is one of the most sensitive regions to anthropogenic and climatic changes mostly affecting its water resources and related practices. With multiple studies raising serious concerns of climate shifts and aridity expansion in the region, this one aims to establish a new high resolution classification for hydrology purposes based on Mediterranean

specific climate indices. This classification is useful in following up hydrological (water resources management, floods, droughts, etc.), and ecohydrological applications such as Mediterranean agriculture like olive cultivation and other environmental practices. The proposed approach includes the use of classic climatic indices and the definition of new climatic indices mainly precipitation seasonality index $I_s$ or evapotranspiration threshold $S_{PET}$ both in line with river flow regimes, a Principal Component Analysis to reduce the number of indices, K-Means classification to distribute them into classes and

finally the construction of a decision tree based on the distances to classes kernels to reproduce the classification without having to repeat the whole process. The classification was set and validated by WorldClim-2 at 1-km high resolution gridded data for the 1970-2000 baseline period and 144 stations data over 30 to 120 years, both at monthly time steps. Climatic classes coincided with a geographical distribution in the Mediterranean ranging from the most seasonal and dry class 1 in the south to the least seasonal and most humid class 5 in the North, showing up the climatic continuity from one place to another and

enhancing the visibility of change trends. The MED-CORDEX ALADIN and CCLM historical and projected data at 12-km and 50-km resolution simulated under RCP 4.5 and 8.5 scenarios for the 2070-2100 period served to assess the climate change impact on this classification by superimposing the projected changes on the baseline grid based classification. RCP scenarios are increasing seasonality index $I_s$ by +80% and aridity index $I_{Arid}$ by +60% in the North and $I_{Arid}$ by + 10% without $I_s$ change in the South, hence causing the wet seasons shortening and river regimes modification with the migration North of winter

moderate and extreme winter regimes instead of early spring regimes. ALADIN and CCLM RCM models have demonstrated an evolution of the Mediterranean region towards arid climate. The classes located to the north are slowly evolving towards moderate coastal classes which might affect hydrologic regimes due to shorter humid seasons and earlier snowmelts. These scenarios might look favourable for Mediterranean cultivation however, the expected impact on water resources and flow regimes will sure expand and directly hit ecosystems, food, health and tourism as risk is interconnected between domains. This

kind of classification might be reproduced at the global scale, using same or other climatic indices specific for each region highlighting their physiographic characteristics and hydrological response.





# 1 Introduction

Mediterranean climate is a result of a complicated global cyclonic system swiping a large evaporative basin. The distribution of marine and continental air masses creates an alternation of low-pressure zones coming over from Iceland and the Persian Gulf or high-pressure zones from Siberia and Azores (Clerget, 1937). The seasonal shifts of these zones are magnified by the North Atlantic Oscillation (NAO) that plays an important role in shaping Mediterranean climate and influencing the evolution of farming and social activities on the long-term (Rodwell and Hoskins, 1996). During the positive phases of the NAO, oceanic

disturbances bring the most humid to northern Europe and less humid to North Africa and the Middle East (Douguédroit & Lionello, 2015). This continuous alternation of high and low pressure, cold and humid winters followed by a hot and dry summers, marks the Mediterranean seasonality which makes the region attractive to social activities, thus its sensitivity to climate change and anthropogenic pressures (PlanBleu, 2012). Hydrologically, the seasonality plays a role in shaping rivers runoff as Haines et al. (1988) classified the Mediterranean rivers under Group 12 Winter Moderate hydrologic regimes, Group

13 Extreme Winter and Group 14 Early Spring, and found a clear relation to the Köppen Csa and Csb climates and a close equivalent of the 'Mediterranean Seasonal' categories of Gentilli (Haines et al., 1988). Seasonality is a main factor in the Mediterranean but to our knowledge its use is still limited as a characterising index for climatic and hydrological classification. Climate change is expected to have severe consequences on Mediterranean runoff with a serious risk of fresh water availability decrease of 2% to 15% for 2°C of warming (Cramer et al., 2018) and significant increase of droughts period particularly in the

South and East (Hreiche et al., 2007; Cudennec et al., 2007; Garcia-Ruiz et al., 2011; Verdier and Viollet, 2015). The CMIP5 simulations (Coupled Model Intercomparison Project, Phase 5) expected a mean precipitation decrease of -4%/°C and temperature increase of 20% more than the global average with maximum precipitation reduction reaching -7%/°C in winter in the southern Mediterranean region and -9%/°C in the summer in the Northern region (Lionello & Scarascia, 2018). At 1.5 °C global warming, some Mediterranean areas are under aridification while moving to drier state due to the decrease in

precipitation combined with PET increase leading to an expansion of drylands, hence affecting more people (Koutroulis, 2019). Automatic classification methods partition a set of objects knowing their distances by pairs in a way to keep the classes as much homogeneous as possible while remaining distinct from each other. Like any classification, the adopted method depends from the objective and its specificity. There are several modes of climatic classification: (a) Genetic classifications related to meteorological causes and the origin of air masses (Bergeron, 1928; Barry and Chorley, 2009). (b) Bioclimatic classifications

based on the interrelation between vegetation type and climate (Holdridge, 1947; Mather and Yoshioka, 1968; Harrison et al., 2010). (c) Agro-climatic method based on the assessment of the Rainfall - Evapotranspiration balance for the estimation of agricultural productivity (Thornthwaite, 1948). (d) Climatic methods based on precipitation and temperature indices similarly





to the classification of Köppen in 1936 (Köppen, 1936) updated by Peel in 2007 (Peel et al., 2007) and which remains the most used.

There are several climate classification studies of the Mediterranean region; among these we cite Köppen-Geiger classification at the global scale in which the Mediterranean climate is well distinctive (Köppen, 1936; Peel et al. 2007; Eveno et al., 2016). Köppen's classification divides the globe into thirty climate zones and relies on a partition hierarchy. It is based on precipitation and temperature indices where Mediterranean climate corresponds to dry hot or dry warm summer where either the precipitation in the driest month in summer is below 40 mm or below the third of the precipitation in the wettest month in

winter (Cs) and the air temperature of the warmest month is above 22 (Csa) or the number of months with air temperature above 10 °C exceeds 4 (Csb). The (Cs) climate doesn't reign all over the Mediterranean region, some exceptions could be observed. A Desertic climate (BWh) dominates Egypt and Libya, (Bsk) Southeast Spain and (Cf) the regions of Thessaloniki and Veneto. On the other hand, and at a global scale, (Cs) climate is present in California, Chile and South Africa (Figure 1). Rivoire et al. (2019) classified 160 Mediterranean rain gauges according to monthly net precipitation (P - ET0). The

classification showed a marked distinction between two clusters with northern stations having a precipitation deficit from April to September and southern stations having a precipitation deficit from March to October. Other climatic classification were also carried out in the Mediterranean but at the national scale like in France, using ascending hierarchical automatic classification based on a 1976 rain gauges network for the 1971-1990 period (Champeaux and Tamburini, 1996); In Turkey seven different climate zones were identified by using Ward's hierarchical cluster analysis based on data from 113 climate

stations for the 1951-1998 period (Unal et al., 2003); Another reclassification of rainfall regions of Turkey was also carried out in 2011 by K-means based on 148 stations covering the 1977-2006 period (Sönmez and Kömüşcü, 2011); We also mention the classification of cyclonic trajectory information using K-means clustering for an 18 years period over the Mediterranean (Trigo et al., 1999); Synoptic meteorology using discriminant analysis over the eastern Mediterranean for 1948–2000 (Alpert et al., 2004); Cloud physical properties classification at the pixel level using K-means applied over European Mediterranean

region (Chéruy and Aires, 2009); The hydrological classification of 40 Mediterranean streams natural flow regimes using PCA to identify the most representing Richter's hydrological indices and agglomerative cluster analysis (Oueslati et al., 2015); However, no specific classification based on precipitation and temperature series has yet treated the Mediterranean region as a climatic or hydrological unit, hence the aim of our study.

The objective of this study is first to establish a Mediterranean specific climatic classification for hydrology purposes based
on a set of indices mainly seasonality and aridity, second, to estimate future evolution of this classification based on Radiative Concentration Pathway (RCP) scenarios with an easy follow up tool using olive cultivation evolution in the Mediterranean.

Through the classification of the Mediterranean catchments climatically and in a second step physiographically, we will be able to characterize their hydrological patterns, useful for the prediction on ungauged basins. This study is a contribution to the HyMeX program and to the Med-CORDEX initiative. The HyMeX program (HYdrological cycle in the Mediterranean

Experiment) aims at a better understanding of the Mediterranean hydrology, with emphasis on the predictability and evolution of decadal variability in the context of global change. Med-CORDEX, a HyMeX initiative, (Ruti et al., 2016), is part of the



COordinated Regional Downscaling EXperiment specific for the Mediterranean that aims at improving our understanding of climate change through high resolution Atmosphere Regional Climate Models (RCM). RCM were introduced in late 1980's as a nested technique into Global Climate Models (GCM) to consider regional scale climatic forcings caused by the complex

physiographic features and small-scale circulation features (Giorgi, 2006). The primary application of RCM has been in the development of climate change scenarios of which we mention ALADIN RCM (Aire Limitée Adaptation dynamique Développement InterNational) developed by Météo France and CCLM (Cosmo Climate Limited-area Model) developed by the German Weather Service (DWD) both applied for EURO-CORDEX and MED-CORDEX projects (Rockel et al., 2008; Tramblay et al., 2013). We aim in this study to discuss the results of of the individual models and not compare their

performances, such study was carried out for EURO-CORDEX with 17 RCM models for the representation of the basic spatiotemporal patterns of the European climate for the period 1989–2008 (Kotlarski et al., 2014).

This paper is structured into six sections; Section 1 Introduction; Section 2 presents the Mediterranean limits and the database; Section 3 the classification approach based on PCA, K-Means and the decision tree with the presentation of MED-CORDEX atmosphere-RCM climate change scenarios; Section 4 the results of WorldClim-2 classification of catchment indices and on

gridded indices and stations, classification projection and impacts under MED-CORDEX scenarios followed by a discussion in Section 5 before concluding in Section 6.

## 2 Study area and database

### 2.1 Defining the Mediterranean region boundaries

From the Latin word Mediterranĕus meaning 'middle land' the Mediterranean refers to the sea and bordering region located in

the middle of the Ecumene between the European, African and Asiatic continents. With Köppen's classification (Köppen, 1936) the definition designated henceforth a moderate climate and extended geographically beyond the limits of the Mediterranean Sea. The question that arises is how would the Mediterranean boundary be defined? Several alternatives are considered in this case based on the field of practice; hydrological boundary was adopted for this study as shown in Figure 1.

\- The climatic boundary could be defined according to Köppen's classification where a set of regions share similar temperature

and precipitation characteristics and known for their warm and dry summers and cold and humid winters. It is limited by the African desert to the South and the temperate European countries to the North.  This boundary might change according to the definition of this similarity. Some regions share a similar Mediterranean climate although located far outside the Ecumene such as Chile, California or South Africa.

\- The hydrological boundary is defined by the set of catchments draining towards the Mediterranean Sea (Milano, 2013). This

definition neglects some of Mediterranean climate regions like Portugal, western Spain and favours geographically adjacent regions like Egypt and Libya.

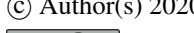

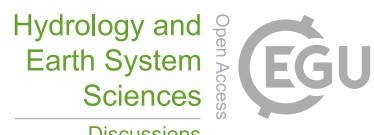

- The agricultural-bioclimatic boundary consists of the set of regions sharing the same types of vegetation considered as indicators of the Mediterranean region such as olives, (Moreno, 2014). This definition is linked to human activity with the same nuances as the climatic limit.

- The administrative boundary of countries adjacent to Mediterranean Sea has a problematic definition independent of any natural base (Wainwright and Thornes, 2004). These boundaries include several climatic classes and cover larger areas than the topographical limits.

## 2.2 Catchments

Since the geographic extent of the study is very wide, the delimitation of catchments was imported from international
references. The European Commission using the Joint Research Centre (JRC) has done extensive and elaborate work on the delimitation of catchments in Europe and some adjacent countries as part of the "Catchment Characterization and Modelling" (CMM) project (de Jager and Vogt, 2010). For catchments in the Middle East and Northern Africa, catchments from HydroSHEDS, the World Wildlife Fund's project, were used (Lehner and Grill, 2013). According to these databases, the total number of catchments extracted at their main stem outlet to Mediterranean coastline and exceeding 1 km$^2$ is 3681 covering a
total area of 1,781,645 km$^2$. It should be noted that the Nile was omitted for its extent 3500 km to the south of the Mediterranean. Catchments surface distribution is shown in Table 1 where middle range catchments, between 100 and 3000 km$^2$, constitute 35% of the total and cover 28% of the total area.

## 2.3 Climatic data

Three types of climatic data were used in this study, (1) WorldClim-2 new 1-km spatial resolution climate surface data (Fick
and Hijmans, 2017), (2) Monthly average times series of 144 stations from NOAA database of 20 different Mediterranean countries covering a period of 30 to 120 years used for validation purpose (3) MED-CORDEX historical and projected data simulated under RCP 4.5 and 8.5 scenarios for future projections. (Tramblay et al., 2013).
(1) WorldClim-2 new 1-km spatial resolution climate surface data, which consists of long-term average monthly temperature and precipitation, solar radiation, vapor pressure and wind speed data, aggregated across a target temporal range of
1970–2000, using data from 9000 to 60000 weather stations (Fick and Hijmans, 2017). Worldclim-2 database is a refined and expanded version of the 2005 "WorldClim-1 database" (Hijmans et al., 2005). This database covers the whole study area, thus climatic classification of Mediterranean catchments was possible. The WorldClim-2 database was built over 23 regions with different coverage for each parameter. For the precipitation an overlap of 3 regions covered the Mediterranean area with a total of 10410 stations for the 3 regions (Western Europe n= 3730; Eastern Europe n = 3632; North Africa n = 3048). For average
temperature, the Mediterranean was covered by one region (eu1) with number of stations n = 1760; n = 1627 for Maximum temperature and n = 1626 for Minimum temperature; Refer to figures S1 and S2 in the supporting information of Fick and Hijmans (2017) article. Monthly precipitation and temperature were averaged for each catchment and then climatic indices calculated at both catchment and grid scale. Climatic characteristics of Mediterranean catchment are summarised and



illustrated in Table 2 and Figure 2, reflecting the wide variability of mean annual precipitation ranging between 5 ("Jabal el
Aswad desert in Libya") and 3000 mm (Kobarid in Slovenia) and mean annual temperature ranging between -14°C (Mont
Blanc, Alps, France) and +26°C (Karak, Jordan) where some catchments receive 50 times more than others the amount of
precipitation while being 4 times colder.

(2) 144 ground weather station data covering the whole study area served to validate the Mediterranean climate classification
with 105 stations located within catchments boundary and 39 outside. Also, 102 of these stations located within Köppen's
(Csa) and (Csb) Mediterranean climate and 42 outside. These stations belong to Global Historical Climatology Network GHCN
(Menne et al., 2012) and recognized by the World Meteorological Organization (WMO), they are available for free access on
the portal of the National Administration of Oceans and Atmosphere of the United States (NOAA). The average length of data
series is 60 years and range between 30 and 120 years at monthly time step. The 1960 - 1990 period is common to all stations.
The data quality was verified (i.e. ellipse of Bois (Bois, 1986)) and only complete hydrological years were retained for indices
calculation.

(3) MED-CORDEX simulations of the Regional Climate Models (RCM) ALADIN-Climate v5.2 at 12 km and CCLM at
50 km spatial resolution grid were used to analyse the climate change impacts on the climatic classification for the end of the
century projection period 2070-2100, and for two different RCP 4.5 and 8.5 scenarios in comparison to the historical 1970-
2000 baseline period (Rockel et al., 2008; Tramblay et al., 2013). We limited the climate change study to ALADIN and CCLM
models since when the article was written those were the only MED-CORDEX models to present the simulation results for
RCP 4.5 and 8.5 for 2070-2100 period with the three required variables available (average temperature (tas), average
precipitation (pr) and average radiation (rlds)).

RCP or Radiative Concentration Pathway is a greenhouse gas (GHG) concentration trajectory adopted by the International
Panel for Climate Change (IPCC) for its fifth Assessment Report (AR5) in 2014. RCP 4.5 and 8.5 were chosen between 4
available scenarios being the most focused on in literature. RCP 4.5 assumes that global annual emissions measured in $CO_2$-
equivalents peak around 2040, with emissions declining substantially thereafter while under RCP 8.5 emissions continue to
rise throughout the 21st century. The RCP 4.5 (resp. RCP 8.5) means that the GHG and aerosols concentrations evolve in a
way that leads to an additional radiative forcing equal to +4.5 W/m$^2$ (resp. + 8.5 W/m$^2$) at the end of the 21$^{st}$ century with
respect to the pre-industrial climate. Consequently, the RCP 4.5 can be considered as an optimist scenario whereas RCP 8.5 is
a more pessimist option. (Giorgi et al., 2009; IPCC, 2013; Ruti et al., 2016).

While  temperature increase and precipitation decrease have been already observed (IPCC, 2013), MED-CORDEX RCP 4.5
scenario projections, as simulated by ALADIN v5.2 for the 2071–2100 period (Tramblay et al., 2013; Dell'Aquila et al. 2018;
Drobinski et al., 2018; Tramblay et al., 2018), estimates a spatially distributed temperature increase of 1.4 to 3.5°C and a
precipitation evolution of ± 10% while RCP 4.5 projects an increase of 2.2 to 6.4°C and a precipitation evolution of ± 20%
compared with the baseline period 1970-2000 with expected shifts of Mediterranean climate and expansion of arid regions
(Beck et al., 2018; Barredo et al., 2019) and related water restrictions and legal decision-making processes (Sauquet et al.,
2018).





The use of ground-based stations time series or gridded observational data is limited by several uncertainties mainly density and interpolation processing methods, especially in Mediterranean region where North African and Levantine countries are poorly covered (Raymond et al., 2016; Zittis, 2018). Nevertheless, the use of specific indices like seasonality and aridity, which are averaged on 30 years periods and based on monthly and annual values, while avoiding extreme event indices, reduces data quality uncertainties. On the other hand, several studies have revealed the uncertainties connected to the resolution of RCM simulated gridded data in the Mediterranean complex domain (Romera et al., 2015) hence the use high-resolution data like MED-CORDEX 12 and 50 km grids and WorldClim-2 1-km and overall, the regional aspect of this study makes it less sensitive to local errors.

### 3 Methodology

The suggested methodology includes first the definition of the climatic indices from which some are classic like the frequency indices and other are specific of the Mediterranean climate like precipitation seasonality. Second, a Principal Component Analysis (PCA) to reduce the number of climate indices and consider only the most contributing. Third, K-Means classification according to the most contributing indices and finally the construction of a Decision Tree based on distances to classes kernels to determine whether or not a place has a Mediterranean climate, and to which type it belongs. This approach was applied at catchment scale, where indices are calculated from averaged climatic variables of each catchment and then verified at grid scale and a set of ground stations. Each class was described and characterized by its corresponding climatic indices. The Mediterranean climatic classes evolution was assessed according to indices variation based on simulated RCP scenarios and by following up the olive tree cultivation boundary as an example of historical Mediterranean specific bioindicator. Olive reproductive cycle displays considerable variations due to climate evolution among other, influencing flowering intensity mainly affected by seasonal temperature and water availability (Moreno, 2014).

### 3.1 Hydrology driven climatic indices

The hydrology driven independent climatic indices were chosen subjectively and developed at the catchment scale from WorldClim-2 monthly average data and divided onto four groups to highlight the Mediterranean seasonality and precipitation intermittence hypothesis of the climate and its corresponding hydrological response. While the flow seasonality is clearly affected by the precipitation seasonality, the other indices help in finetuning this theory like monthly temperature and potential evapotranspiration variation. A complete list of indices with a description of each is in Table 3.

Group I: indices based on monthly precipitation from which we mention seasonality index $I_S$, peak indices $P_{1.5}$, $P_2$ and frequency indices $P_{25\%}$, $P_{75\%}$. $I_S$ is directly linked to Mediterranean flow regimes for expressing the precipitation ratio between the 3 most humid months and the 3 most dry months with values ranging from 0 to 1 (Hreiche, 2003). $I_S$ values tending towards 0 express uniform distribution of precipitation along the year with a hydrological response lacking flood and drought seasons





while $I_S$ values tending towards 1 correspond to a normal distribution of precipitation with a hydrological response more likely to show flood and drought seasons.

Group II: indices based on monthly temperature expressed by the temperature lag between the coldest and warmest months $\Delta T_1$, frequency indices $T_{25\%}$, the number of months exceeding the average Mediterranean temperature $S_{Tm}$.

Group III: indices based on both temperature and precipitation expressed by $I_{Decal}$ the time lag between the coldest and most humid month.

Group IV: indices based on precipitation and evapotranspiration expressed by aridity index $I_{Arid}$. The evapotranspiration was

estimated according to Turc's formula (Turc, 1961), chosen for its application simplicity and adequacy to Mediterranean areas as it was originally developed for southern France and North African countries (Diouf, 2016). Turc's formula is mainly based on temperature and radiation, two stable parameters on the regional scale which reduces the uncertainties when using regionalized dataset such as WorldClim-2. Group II and IV indices describe the seasonality and variability of evapotranspiration and intermittence of wet and dry seasons.

**3.2 Principle Component Analysis**

Principal Component Analysis (PCA) is widely applied to reduce the dimensionality of datasets and keeping the most representing and uncorrelated variables. This section presents a brief description of the method along with some of their applications in hydrology. For an extensive mathematical description and demonstration of these methods we advise to consult; Krzanowski (1988) and Jollife (2002).

PCA was first introduced by Karl Pearson (Pearson, 1901) and then developed by Harold Hotelling (Hotelling 1933). Hotelling's motivation is that there may be a smaller *fundamental set of independent variables which determine the values* and conserve the maximum amount of information of the original variables (Jolliffe, 2002). This is achieved by transforming a vector of *p* random variables to a new set of variables, named Principal Components (PC), by looking for a linear function of the elements having maximum variance. And next looking for another linear function uncorrelated with the first and having

maximum variance and so on up to *p* PCs. It is hoped in general, that most of the variation will be accounted for by *m* PCs, where *m < p*.

The number of indices is reduced at two steps. The first step is based on the correlation matrix, where strongly correlated indices higher were eliminated (a threshold of 0.85 was chosen for this study). The second is based on PCA results where indices that doesn't contribute into the principal component that represent the greatest variabilities are eliminated.

**3.3 K-means clustering technique**

Cluster analysis consists of data points partitioning into isolated groups while minimizing the distance between same cluster data points and maximizing it between different clusters. One of the most popular clustering methods is the K-means method introduced by Edward Forgy (Forgy, 1965) and MacQueen (MacQueen, 1967). It aims to minimize the square error objective function for distance optimization. The optimization steps begin with (1) kernels initialization, the kernel being a virtual point



representing the statistical centre of a class, (2) updating classes, (3) re-evaluation of kernels and (4) repetition of steps (2) and (3) until stabilization. The quality of the solution thus found strongly depends on the initial kernels. In its turn, kernel initialization is sensitive to the data dimensionality. Classification gives a deterministic result where each point should belong to one of the classes, a result of a set of decision rules based on its distances to classes kernels.

The application of K-means requires setting a number of classes, otherwise the optimization leads to as many classes as

individuals. The optimum number of classes 'K' could be defined according to Elbow method (Bholowalia & Kumar, 2014). K-means gained in reputation the last decades and was widely applied in hydrology field for clouds classification from satellite imagery (Desbois et al., 1982), for climatic classification using measured and simulated timeseries (Moron et al., 2008; Carvalho et al. 2016) for catchment classification based on streamflow characterization and precipitation (Toth, 2013). K-Means classification was applied, and catchments were distributed into 5 classes kernels to determine whether they belong,

or not, to a Mediterranean climate and to which type they belong to, if so. We hoped for a classification that delimits the Mediterranean climate from South and North and divides the intermediate coastal zone. Therefore, a distribution into 5 classes was chosen despite that 3 classes would be optimal as per the elbow method. In detail, one class that covers the southern desertic region, another class that covers the northern continental region of non-Mediterranean climate and 3 classes that cover the intermediate coastal region. A larger number of classes would produce an uninterpretable fragmented classification.

**3.4 Decision Tree**

A decision tree is a set of distance criteria or questions in the form of hierarchy that leads to an intended classification (Breiman 1984). To classify new points, stations or reproduce the classification on another dataset, it suffices to define the distance criterion to the various kernels of the climatic classes by predicting values of a dependent variable based on values of predictor variables from a reference classification. This procedure provides validation tools for exploratory and confirmatory

classification analysis.

We generated a decision tree based on the distances to the clusters' kernels obtained from the gridded indices classification. The aim of this decision tree is to easily reproduce the classification with same kernels rather than repeat the whole classification process which will modify the clusters and their kernels. By conserving the same kernels, the decision tree will permit to follow up the climate evolution and its impact on the classification under other scenarios.

In our case, the dependent variables are the climatic classes obtained from K-Means clustering while the predictor variables are the distances to each clusters' kernels. This procedure was done for both catchments and gridded classification. The decision tree generates a set of classification rules usually used to classify new stations based on their distances to classes kernels. In this study, these rules were used in section 5 to classify the projected indices. This way have fixed the classes kernels indices of the 1970-2000 baseline period and calculated the distances of the 2070-2100 projected grid to baseline to

compare both the classification indices and spatial evolution.





### 3.5 RCP Scenarios

For climate change impact assessment, temperature and precipitation delta change were calculated between both baseline period 1970-2000 and projected period 2070-2100 for MED-CORDEX RCM ALADIN and RCM CCLM grids and for two different RCP scenarios (RCP 4.5 and RCP 8.5). Those delta changes were then superimposed to the WorldClim-2 grid, based

on the nearest Euclidean distance between MED-CORDEX grid cells and WorldClim-2 grid cells using GIS spatial join toolbox. The indices were then recalculated using the projected values of monthly temperatures and precipitation. The decision tree rules from Table 6 were then applied for the projected period and the climate change under RCP was illustrated in Figure 7 and expressed by indices evolution between classes in Table 7.

### 3.6 Adopted Methodology

The proposed methodology consists on calculating the climatic indices using WorldClim-2 monthly gridded data averaged at the catchment scale using ArcGIS zonal statistics. The climatic indices were PCA-reduced and classified using K-Means clustering. The classification was verified and compared to WorldClim-2 gridded indices and ground stations indices. In addition to the construction of a hierarchical decision tree to classify projected indices and to avoid repeating the whole process. All PCA, K-Means and the decision tree where calculated using SPSS software. Projected indices under RCP scenarios were

calculated and classification evolution were then deduced.

## 4 Results

This section details the climatic indices derived from the collected database, the results of PCA/K-means classification of each set of indices and their validation on gridded and station indices with a decision tree for replicating the classification on new stations or grids.

**4.1 PCA results for WorldClim-2 catchment based indices**

The number of indices was reduced the first time based on the correlation matrix and the second based on PCA results. We eliminated the strongly correlated indices (correlation higher than 0.85) and 11 indices were kept upon the first step.

- $I_s$ and $P_{75\%}$ are strongly inversely correlated (-0.959). $I_s$ was kept.
- $\Delta T_1$ and $\Delta T_2$ are strongly correlated (0.989). $\Delta T_1$ was kept.

- $T_{25\%}$, $T_{75\%}$ and $D_j$ are strongly inversely correlated (-0.954 and 0.874). $T_{25\%}$ was kept.
- $P_{25\%}$ and $I_{Hor}$ are strongly correlated (0.858). $P_{25\%}$ was kept.

Once the correlation matrix transformed into a diagonal one, it was possible to find the eigenvalues representing the projection from p to k dimensions. The eigenvector matrix is the linear expression of the indices with respect to the principal components. The first eigenvalue 6.36 represents 58% of the variability and the second 1.31 represents 12%. The first two factors F1 and



F2 represent the two greatest variabilities with respect to the following factors and 70 % of the total variability is thus preserved with this choice. Upon the PCA, the number of indices was reduced to 7 showing that $I_s$, $P_{25\%}$, $SP_{1.5}$, $I_{Arid}$, $T_{25\%}$, $S_{PET}$ and $S_{Tm}$ were the most contributing climatic indices with 70% of total variance explained for the first two components. Statistical summaries are shown in Table 4 with $I_S$ values ranging between 0.2 and 1 with an average of 0.8 highlighting Mediterranean seasonality.

**4.2 Catchment based classification**

The K-Means classification shows in Figure 4 a distribution into 5 classes where:

- Class 1: present between Egypt and Libya, highlighting a desertic influence with few rain episodes registered per year, if any, expressed by $I_s = 0.99$ and $I_{Arid} = 11.7$ on average. Precipitation never exceeds evapotranspiration in this region, hence $S_{PET} = 0$.
- Class 2: mainly present in the south and east of the Mediterranean, characterised by a high seasonality $I_s = 0.95$ and high aridity $I_{Arid} = 4.3$.
- Class 3: dominates the central region from the southern tip of Spain to Syria with an average seasonality $I_s = 0.91$.
- Class 4: covers mainly coastal catchments in north-west countries, south-east Italy, western Greece and present discontinuously in the south-west. $I_s = 0.71$ in this class.
- Class 5 only present in northern non-coastal catchments and characterised by a low seasonality $I_s = 0.47$.

In comparison to Köppen's, classes 2, 3 and 4 matches with (Csa) while classes 1 and 5 are mainly outside (Csa) and (Csb) henceforth defined as Non-Mediterranean climate. The main difference with Köppen's Mediterranean classes resides in Southern Spain defined as arid climate (Bsk) while in the present classification it varies between classes 2, 3 and 4. This new distribution indicates climate variability within (Csa) or (Csb), hence the importance of a fine gridded classification. This
variability is highlighted in the class kernels indices (Figure 3) and is mainly due to the complex seasonality across the Mediterranean. This complexity is shown here more delicately than the one defined by Köppen which is climate oriented only and limited to the simple criteria of a wet winter and dry or temperate summer. Therefore, we think that a hydrology oriented climatic classification should account for an intra climate characteristics expressed by specific indices like the one shown here, specific to the Mediterranean and expressed by $I_s$.

**4.3 Grid based classification**

The K-Means clustering of WoldClim-2 gridded data resulted with a spatial distribution similar to the catchment indices classification where class 1 dominates the south, class 5 the north and classes 2, 3 and 4 the central region (Figure 5). This classification has shown better resolution and revealed the shifts of some regions to adjacent classes. Class 4 climate appeared on Spanish coasts, class 3 climate appeared on Sardinia and Greece, Class 2 in Syria and a limited spread of class 4 and 5 on





Eastern Turkey. However, climate continuity is conserved in this classification for indices are gradually increasing or decreasing from North to South.

We believe that this classification is useful both for hydrological and ecohydrological applications like cultivation and other related environmental practices affected by water resources and river flows. Olive is one of the best Mediterranean-specific physiographic indices and we noticed that its cultivation boundary is limited by those of classes 1 and 5 where 13% is in Class

2, 49% in class 3 and 34% in class 4. This observation gives an accurate idea of suitable climate conditions for olive cultivation, deducing that extreme seasonality combined with very high aridity (South) or very low seasonality combined with high humidity (North) are avoided by olive trees. In a similar way, other tree types like pine trees also characterise Mediterranean landscape putting forward the need for a physiographic classification to interpret in parallel to this climatic classification under the umbrella of hydrological characterisation. The future of Mediterranean cultivation in case of climate change is to be

checked under RCP 4.5 and 8.5 scenarios in next section.

### 4.4 Verification on stations indices

The 144 stations were also K-Means clustered based on the selected indices from the PCA. The resulting geographical distribution differed only by some shifting due to averaging and normalization as the sample is much less than the gridded cells. There is no coverage of class 1 as no weather station was found in that region (Figure 6). Despite the shifting, there is an

82% accuracy rate or 86 out of 105 stations located within catchments boundary that matched the gridded distribution, the rest is located within the adjacent classes boundaries. As for olive boundary, there was only one class 5 station corresponding to Firenze that was located within the boundary.

### 4.5 Decision tree analysis

A decision tree was generated based on the gridded indices and their distances to cluster's kernels. The total population of

gridded indices was divided randomly into two equal subsets, one for training and the second for testing. The predicted classes values of both sets were then compared to the original classification of the gridded indices obtained in section 4.3 and both yielded an overall 93% accuracy (Table 5). We notice that some grids have joined one of the adjacent classes due to interclass connectivity; this confirms once more the continuity of climate. The generated decision tree of 3 levels includes 75 nodes in total due to high population number with 75 classification rules sampled in (Table 6). As an example, for class 1, if the distance

to kernel 1 (D1) is below 3.5 and the distance to kernel 2 (D2) is above 2.2, then the grid cell belongs to class 1. This decision tree permits to follow up the climate evolution and its impact on the classification applied in section 5.

### 4.6 MED-CORDEX ALADIN RCP Scenarios Climate Evolution

Under RCP 4.5 scenario, temperature is increasing by 1.4 to 3.5°C (average 2.2°C), with the lowest rates during winter and the highest during summer. In the South, on average, precipitation is increasing by 25% during winter and 70% during summer





and decreasing by 15% during spring and 5% during fall. In the North it is increasing by 10% during winter, spring and fall
       while staying stable along the year in the central region. No major area changes are occurring between classes. In detail, class
       5 is reducing its extent in Greece and Albania in favour of classes 3 and 4 but compensating in central Spain; class 3 extent is
       decreasing in Turkey and Corsica in favour of class 4 in Lebanon and class 2 in Cyprus. Classes 1, 2 and 3 seasonality indices
       $I_S$ are stable while classes 4 and 5 are increasing by 7% and 9%. Also for classes 4 and 5, $S_{P1.5}$ is highly increasing (70%) with

$P_{25\%}$ staying almost the same (3%) which means that precipitation change is temporally distributed in a way that more months
       are exceeding the average monthly precipitation by 1.5 times and that the humid season has shortened, enhancing seasonality
       variation. Another remarkable change is class 5 $I_{Arid}$ 20% increase pushing it towards class 4.
       Under RCP 8.5 scenario, the case is accentuated for temperature which is increasing evenly across the Mediterranean by 2.5
       to 5.6°C (average 3.8°C) with the lowest rates during winter and the highest during summer. In the South, on average,

precipitation is increasing by 60% during summer and decreasing by 10% during winter. In the North it is increasing by 5%
       during spring and summer while staying almost stable along the year in the central region. The area also didn't change much
       under RCP 8.5; in detail class 3 is taking over the south eastern coast of Spain but retreating in favour of class 4 from North
       West Africa and Turkey. The difference with RCP 4.5 scenario resides first in the indices evolution where $I_s$ is increasing by
       9% in class 5 and $S_{P1.5}$ highly increasing by 96%. This has caused an area change of 2% towards class 4 mainly in Spain,

Greece and Albania. Another change is class 3 $I_{Arid}$ increasing by 19% and $S_{PET}$ decreasing by 10% which means that this
       moderate region is pushing towards more arid climate

## 4.7 MED-CORDEX CCLM RCP Scenarios Climate Evolution

       Under RCP 4.5 scenario, temperature is increasing by 1.9 to 3.5°C (average 2.9°C), with the lowest rates in the South during
       winter and the highest in the North during summer. In the South, on average, precipitation is increasing by 20% during winter

and 10% during fall and decreasing by 10% during summer but stable during spring. In the North it is increasing up to 10%
       during fall and winter and decreasing down to 30% during spring and summer. The spatial extent of class 5 is increasing by
       4 % mostly in northern Spain, Albania, Morocco and Algeria in favour of class 4 which is decreasing by 5%; Class 3 appeared
       between Italy and France on the Ligurian Sea, at San Marino and on the Spanish coast; Class 2 expanding over the Turkish
       coast and in Morocco; Class 1 remained almost unchanged. Classes 1 and 2 seasonality indices $I_S$ are constant while of classes

3, 4 and 5 are increasing by 4%, 27% and 42%. Also, $S_{P1.5}$ is increasing by 14%, 120% and 320% for classes 3, 4 and 5 with
       a little change of $P_{25\%}$ (less than 10%); same observation as ALADIN RCP4.5 scenario but more accented. $I_{Arid}$ is increasing
       by 56% in class 5 while $S_{PET}$ is increasing by 25%.
       Under RCP 8.5 scenario, temperature is increasing by 3.6 to 6.4°C (average 5.1°C), with the lowest rates in the South during
       winter and the highest in the North during summer. In the South, on average, precipitation is increasing by 30% during winter

and 10% during fall and decreasing down to 25% during summer but stable during spring. In the North it is increasing up to
       10% during fall and winter and decreasing down to 60% during spring and summer. Spatially, class 3 is increasing by 9%





mostly in Italy, France, Spain, northern Greece and Algeria in favour of class 4 which is decreasing by 7%; Classes 1, 2 and 5 remained almost unchanged. Classes 3, 4 and 5 seasonality indices $I_S$ are increasing by 8%, 39% and 80%. $S_{P1.5}$ is increasing by17%, 215% and 516% for classes 3, 4 and 5 with a change of $P_{25\%}$ of maximum 17%. Aridity indices are increasing of 24%

to 50% for classes 3, 4 and 5 while $S_{PET}$ is decreasing between 13 and 26 % for the same classes. In summary, the Mediterranean is evolving towards an arid region under both CCLM RCP 4.5 and 8.5 scenarios. The climate change under RCP was illustrated in Figure 7 and expressed by indices evolution between classes in Table 7.

## 5 Discussion

The objective of this study is first to establish a Mediterranean specific climatic classification for hydrology purposes based

on a set of indices mainly seasonality and aridity, second, to estimate future evolution of this classification based on RCP scenarios.

The continuous evolution of climate across the Mediterranean was demonstrated by the indices values uniformly increasing or decreasing from North to South in all classifications (Figures 4, 5, and 6). Seasonality is highest in the South and lowest in North, same for other precipitation indices and aridity. The catchment based classification put the whole catchment within the

same class despite the intra climatic diversity which mostly affected wide catchments (above 10000 km$^2$) like Rhône, Ebro and Po and to less extent, smaller catchments (less than 3000 km$^2$) as climatic diversity decreases with area and spatial spread. It is interesting to cross analyse this classification with a catchment based physiographic classification (article in preparation) which both classifications will be used for a hydrological characterization of Mediterranean catchments. The grid based classification refined the catchment based classification showing different climatic classes within the same catchment mainly

between coastal low land areas, valleys and mountainous high land areas. However, we could still notice in Figure 5 that Alps mountains and Po valley are still in the same class according to our classification approach as they both share close seasonality index ($I_s \approx 0.47$) and aridity index ($I_{Arid} \approx 1.06$). In general, the classification gradient was conserved from class 1 in the South to class 5 in the North which confirms that $I_s$ and $I_{Arid}$ are the main contributors to the classification taking over precipitation and temperature frequency indices. Nevertheless, class 3 spots were seen in Northern Italy mountains at the boundary with

Austria which upon checking appeared to have a higher seasonality and aridity ($I_s \approx 0.70$; $I_{Arid} \approx 0.98$) than the surrounding region ($I_s \approx 0.61$; $I_{Arid} \approx 0.78$), an anomaly that might be caused by variables interpolations in the area.

In the North, where seasonality is low and precipitation is regular along the year, RCP 4.5 and 8.5 scenarios impacts on hydrology are more accentuated for CCLM than ALADIN as the first is projecting a high precipitation decrease, down to -30% and -60% and a warming of 3.8°C and 6.8°C for RCP 4.5 and 8.5 consequently during dry spring and summer seasons,

hence increasing $I_s$ by +80% and $I_{Arid}$ by +60% causing the wet seasons shortening and river regimes modification with the migration North of Group 12 Winter Moderate regimes instead of Group 14 Early Spring regimes. ALADIN is projecting a moderate precipitation variation of ± 10% with a warming of 2.7 °C and 4.5°C and increasing $I_s$ by only +9% and $I_{Arid}$ by +20% (see Table 7).


In the South, where seasonality is very high already, and precipitation is limited to fall and winter, models have projected little
to no modification. RCP 4.5 and 8.5 scenarios impacts on hydrology is more accentuated for ALADIN than CCLM as the first
is projecting a precipitation change between -5% and +25% for RCP 4.5 and between -12% and -2% for RCP 8.5 during fall
and winter consequently with $I_{Arid}$ change reaching 10%; CCLM is projecting a precipitation increase between +8% and +22%
for RCP 4.5 and between +5% and +32% for RCP 8.5 with only +3% $I_{Arid}$ change while $I_s$ didn't change for both. A
modification of hydrologic regime from Group 14 Early Spring to Group 13 Extreme Winter is expected.

Looking to the maps in Figure 7 we can easily notice that class 2 and 3 are expanding to the North for RCM CCLM while this
change is limited for RCM ALADIN; looking South, we don't see much change on the maps, hence confirming our previous
observations.

The use of ALADIN and CCLM models is not enough to fully assess the uncertainties which is beyond the scope of this paper.
Nevertheless, the seasonal variability between models and scenarios, despite the general trend towards warming, aridity and
accentuated seasonality, incited us to address the main reasons behind. This uncertainty usually depends on adopted climate
variables, the region, seasons (Lionello & Scarascia, 2018). In addition, the adopted models in this study are atmosphere-RCM
and not fully coupled models, as they are not yet achieved by the MED-CORDEX, which could have returned different results.
ALADIN and CCLM RCM models have demonstrated an evolution of the Mediterranean region towards arid climate, more
emphasized with CCLM especially for RCP 8.5. These scenarios might look Mediterranean friendly as classes 4 and 5
seasonality indices are evolving towards class 3 in addition to some spatial expansion which might look favourable for
Mediterranean cultivation however, the expected impact on water resources and flow regimes will sure expand and directly
hit ecosystems, food, health and tourism as risk is interconnected between domains (Cramer et al., 2018).

## 6 Conclusion

The Mediterranean climate characteristics and specifically precipitation seasonality, main contributor according to PCA, plays
an important role in the hydrological mechanisms of Mediterranean catchments and flow intermittence. A decision tree makes
it possible to define, from distances to class kernels, if any place has a Mediterranean climate or not, and to which type of
Mediterranean climate does it belong to, for present and future scenarios. On the other hand, the superposition of olive
cultivation boundary as Mediterranean-specific physiographic index highlighted the utility and importance of physiographic-
climatic coupled scenario models that could be extended to other Mediterranean physiographic or bio-climatic indices. The
climatic classification and corresponding indices evolution under RCP scenarios helped in identifying the general climate
change impact on Mediterranean seasonality that might uncover valuable findings about water balance, floods and droughts
for water sector stakeholders. Both ALADIN and CCLM scenarios showed an increase of the average seasonality and aridity
indices affecting hydrologic regimes due to shorter humid seasons and earlier snowmelts. The results of this study are useful
for future water resources and cultivation management policies to identify the most impacted zones and propose preventive
and adaptative measures for a more resilient and sustainable region. This kind of classification might be reproduced at the



global scale, using same or other region-specific climatic indices highlighting their physiographic characteristics and hydrological response.

## Acknowledgments

This work is a contribution to the HyMeX program (HYdrological cycle in the Mediterranean Experiment) through INSU-
MISTRALS support and to the Med-CORDEX initiative (COordinated Regional climate Downscaling EXperiment – Mediterranean region). The ALADIN simulations used in the current work were downloaded from the Med-CORDEX database (www.medcordex.eu).

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





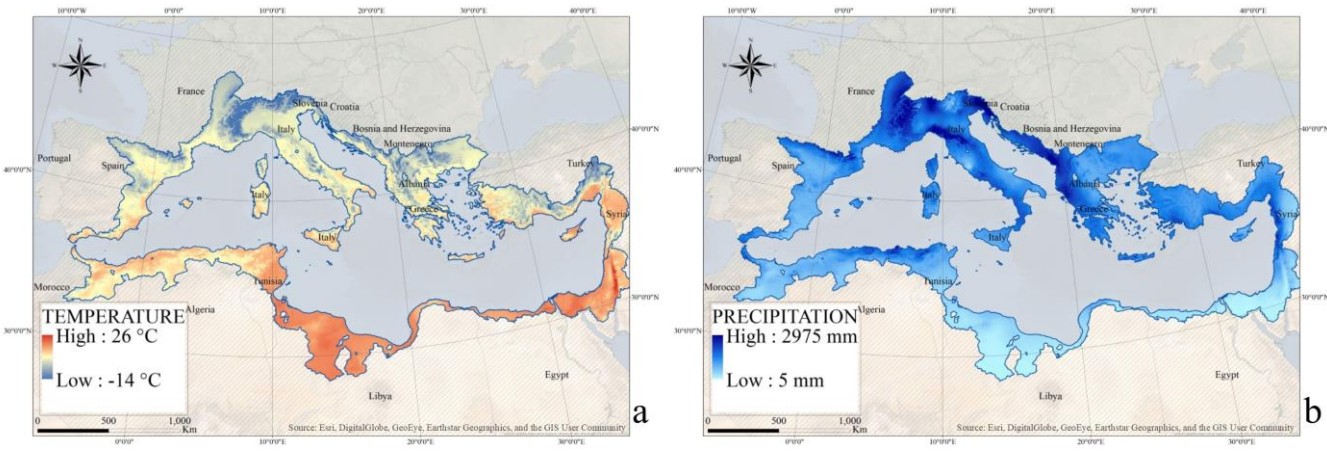

**Figure 1 Four Mediterranean region boundaries (Merheb et al. 2016); first administrative, second topographic (Milano 2013), third olive cultivation (Moreno 2014) and fourth climatic (Peel et al. 2007)**

**Figure 2 WorldClim-2 gridded mean annual temperature in °C (a) and mean annual precipitation in mm (b) from (Fick and Hijmans 2017)**





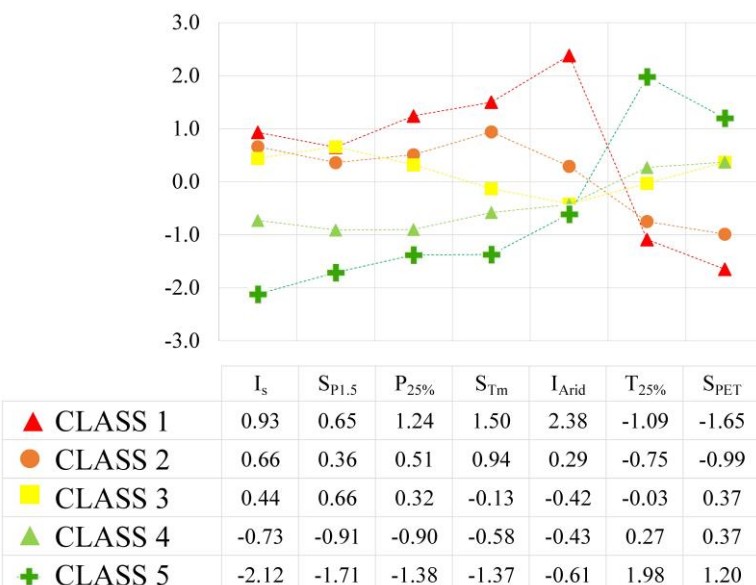

| | $I_s$ | $S_{P1.5}$ | $P_{25\%}$ | $S_{Tm}$ | $I_{Arid}$ | $T_{25\%}$ | $S_{PET}$ |
|---|---|---|---|---|---|---|---|
| ▲ CLASS 1 | 0.93 | 0.65 | 1.24 | 1.50 | 2.38 | -1.09 | -1.65 |
| ● CLASS 2 | 0.66 | 0.36 | 0.51 | 0.94 | 0.29 | -0.75 | -0.99 |
| ■ CLASS 3 | 0.44 | 0.66 | 0.32 | -0.13 | -0.42 | -0.03 | 0.37 |
| ▲ CLASS 4 | -0.73 | -0.91 | -0.90 | -0.58 | -0.43 | 0.27 | 0.37 |
| ✚ CLASS 5 | -2.12 | -1.71 | -1.38 | -1.37 | -0.61 | 1.98 | 1.20 |

**Figure 3: Normalized indices values of the five climatic classes kernels from the Mediterranean catchment's classification using WorldClim-2 data**

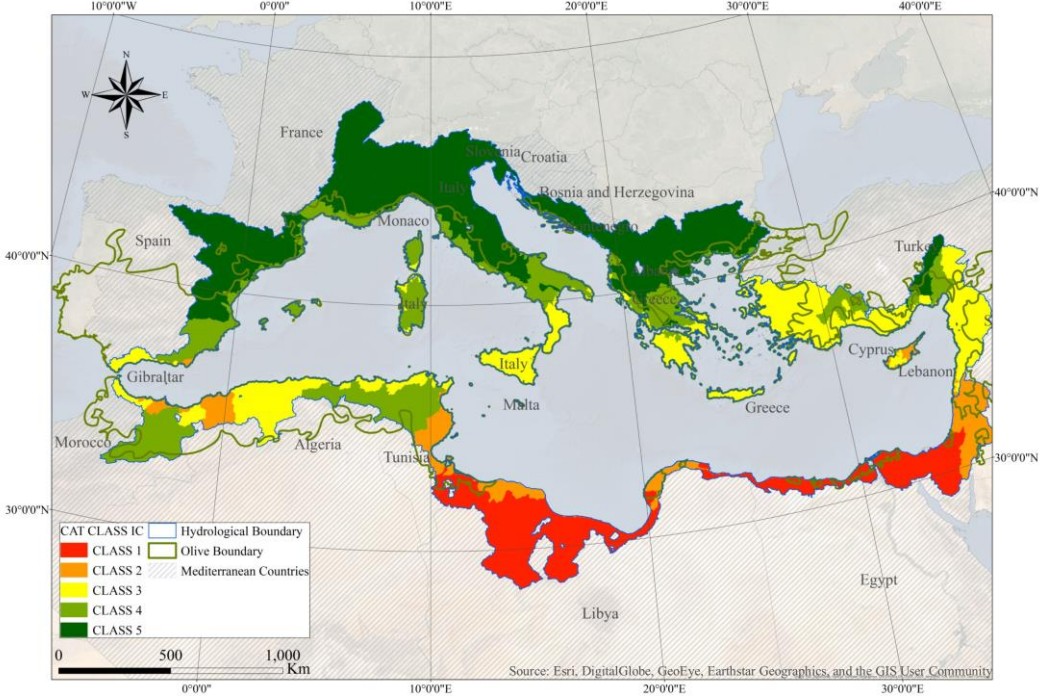

**Figure 4. Geographical distribution of the Mediterranean climatic classes based on average catchments indices using WorldClim-2 monthly data.**





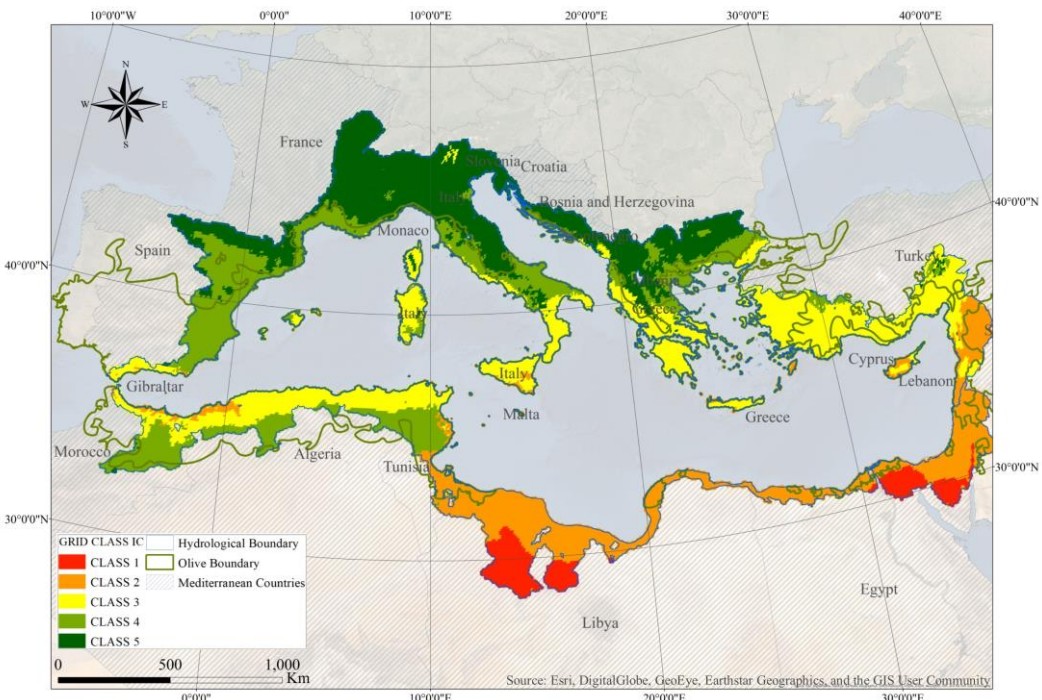

**Figure 5. Geographical distribution of the Mediterranean climatic classes based on gridded indices using WorldClim-2 monthly data**

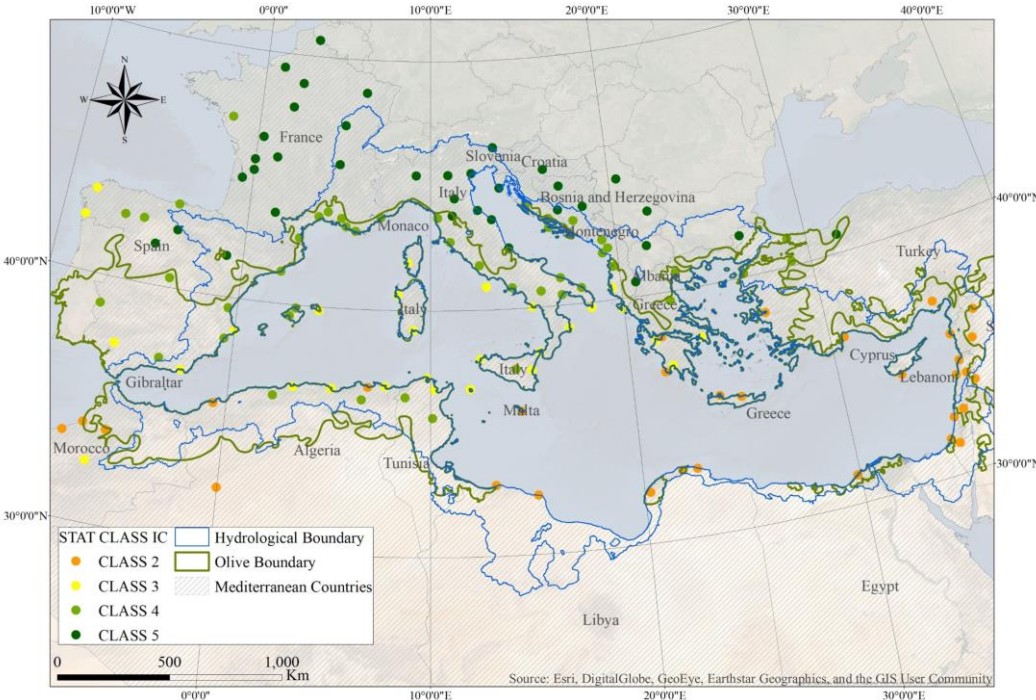

**Figure 6. Geographical distribution of the Mediterranean climatic classes based on 144 stations climatic indices**

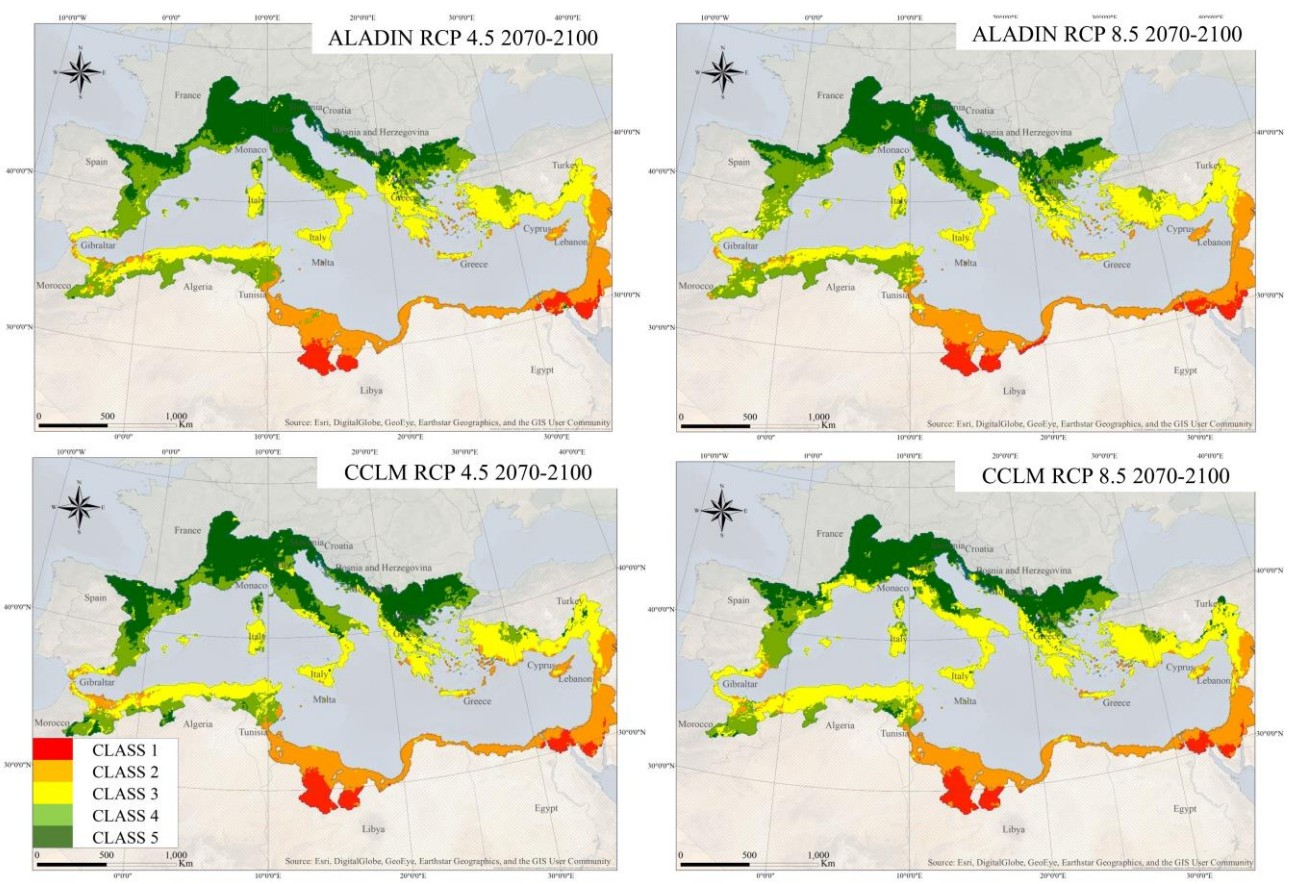

**Figure 7. Projected geographical distribution of the Mediterranean climatic classes based on WorldClim-2 gridded climatic indices using projected data under Aladin and CCLM RCP 4.5 and 8.5 scenarios for the 2070-2100 period.**



| Area range | Number of Catchments | Number of Catchments Ratio | Total Area (km$^2$) | Ratio Area |
|---|---|---|---|---|
| A < 100km$^2$ | 2333 | 63% | 80,157 | 4% |
| 100 km$^2$ < A < 3000 km$^2$ | 1270 | 35% | 498,614 | 28% |
| A > 3000 km$^2$ | 78 | 2% | 1,202,874 | 68% |

**Table 1 Catchment distribution per area and ratio to total area**

| | Area (km$^2$) | $Z_{Max}$ (m) | $Z_{Mean}$ (m) | MAP (mm) | MAT (°C) | MPET (mm) |
|---|---|---|---|---|---|---|
| Minimum | 0.01 | -2 | -4 | 39 | 5.1 | 444 |
| Mean | 467.5 | 737 | 255 | 595 | 16.5 | 1136 |
| Maximum | 96619.0 | 4783 | 1727 | 2004 | 21.7 | 1498 |
| Median | 54.4 | 598 | 185 | 592 | 16.5 | 1127 |

**Table 2 Statistical summaries for the catchments climatic parameters (Maximum altitude ($Z_{Max}$), Mean Altitude ($Z_{Mean}$), Mean Annual Precipitation (MAP), Mean Annual Temperature (MAT), Mean Potential Evapotranspiration (MPET))**

| GROUP | TYPE | CLIMATIC INDICES | DESCRIPTION |
|---|---|---|---|
| I | Climatic Indices based on average monthly rainfall | Seasonality Index $I_S$ | One minus the precipitation ratio between the most three dry humid and dry months |
| | | Precipitation Index $P_{25\%}$ and $P_{75\%}$ | Rain value exceeded 25% or 75% of the time |
| | | Peak index $S_{P1.5}$ $S_{P1.7}$ $S_{P2}$ | Number of months exceeding the average monthly precipitation by 1.5, 1.7 and 2 times |
| | | Horizontal Inertia Index $I_{Hor}$ | Dispersion of monthly rainfall compared to the annual average |
| II | Climatic Indices based on average monthly temperature | $\Delta T_1$ | Temperature lag between the coldest and warmest months |
| | | $\Delta T_2$ | Temperature lag between the coldest and warmest three consecutive months |
| | | Temperature Index $T_{25\%}$ and $T_{75\%}$ | Temperature value exceeded 25% and 75% of the time |
| | | Peak Index $S_{T1.2}$ | Number of months exceeding the average temperature by 1.2 times |
| | | Degree Day $D_j$ | Decomposition according to the need for habitat heating |
| | | Mean temperature Index $S_{Tm}$ | Number of months exceeding the Mediterranean monthly average temperature Tm 16.4 °C |
| III | Climatic Indices based on precipitation and temperature | Time Lag Index $I_{Decal}$ | Time lag between the coldest and most humid month |
| IV | Climatic indices of Evapotranspiration | Aridity Index $I_{Arid}$ | Annual evapotranspiration over annual precipitation $I_{Arid} = PET/P$ |
| | | Threshold Index $S_{PET}$ | Number of month where precipitation P exceeds evapotranspiration PET, (PET calculated using Turc (1961) formula) |

**Table 3 Climatic Indices definition**





|  | $I_s$ | $S_{P1.5}$ | $P_{25\%}$ | $S_{Tm}$ | $I_{Arid}$ | $T_{25\%}$ | $S_{PET}$ |
|---|---|---|---|---|---|---|---|
| Minimum | 0.2 | 0.0 | 0.0 | 0.0 | 0.3 | 1.2 | 0.0 |
| Mean | 0.8 | 2.7 | 0.3 | 5.9 | 3.2 | 1.4 | 4.0 |
| Maximum | 1.0 | 5.0 | 0.9 | 10.0 | 38.3 | 2.3 | 12.0 |
| Median | 0.9 | 3.0 | 0.3 | 6.0 | 1.8 | 1.3 | 5.0 |

**Table 4 Statistical summaries of the PCA selected climatic indices average for catchments using WorldClim-2 monthly data**

| | Sample | 1 | 2 | 3 | 4 | 5 | Accuracy |
|---|---|---|---|---|---|---|---|
| **Training** | 1 | 636 | 26 | 1 | 1 | 2 | 95.5% |
| | 2 | 86 | 1915 | 131 | 0 | 0 | 89.8% |
| | 3 | 0 | 118 | 3537 | 186 | 17 | 91.7% |
| | 4 | 1 | 0 | 135 | 2860 | 68 | 93.3% |
| | 5 | 0 | 0 | 1 | 72 | 3511 | 98.0% |
| | Overall Percentage | 5.4% | 15.5% | 28.6% | 23.4% | 27.0% | 93.6% |
| **Test** | 1 | 637 | 33 | 2 | 2 | 0 | 94.5% |
| | 2 | 71 | 1889 | 166 | 0 | 0 | 88.9% |
| | 3 | 1 | 124 | 3635 | 197 | 11 | 91.6% |
| | 4 | 0 | 0 | 167 | 2912 | 69 | 92.5% |
| | 5 | 0 | 0 | 0 | 83 | 3389 | 97.6% |
| | Overall Percentage | 5.3% | 15.3% | 29.7% | 23.9% | 25.9% | 93.1% |

**Table 5 Gridded classification decision tree accuracy table. The accuracy rate is calculated in comparison to the K-Means classification of the gridded indices in section 4.4.1**

| CLASS 1 (4 rules) | (D1) < 3.5 and (D2) > 2.2 |
|---|---|
| CLASS 2 (13 rules) | (D1) < 3.5 and 1.9 < (D2) < 2.2<br>3.5 < (D1) < 4.2 and 2.4 < (D4) < 2.8 and (D2) < 2.2<br>3.5 < (D1) < 4.2 and 2.8 < (D4) < 3.4<br>4.7 < (D1) < 4.8 and (D4) > 3.4<br>4.8 < (D1) < 5.1 and (D4) > 3.4 |
| CLASS 3 (23 rules) | 3.5 < (D1) < 4.2 and 1.8 < (D4) < 2.1 and (D2) < 2.2<br>3.5 < (D1) < 4.2 and 2.1 < (D4) < 2.4<br>5.1 < (D1) < 5.5 and 1.5 < (D4) < 1.8 and (D5) > 1.7<br>5.1 < (D1) < 5.5 and 1.5 < (D4) > 1.8 |
| CLASS 4 (23 rules) | 3.5 < (D1) < 4.2 and (D4) < 1.8<br>3.5 < (D1) < 4.2 and 1.8 < (D4) < 2.1 and (D2) > 2.2<br>5.5 < (D1) < 5.9 and 1.3 < (D5) < 1.7 and 1.2 < (D4) < 1.5<br>5.5 < (D1) < 5.9 and (D5) > 1.7 |
| CLASS 5 (12 rules) | 5.1 < (D1) < 5.5 and 1 < (D4) < 1.2 and (D5) < 1.3<br>5.1 < (D1) < 5.5 and 1.5 < (D4) < 1.5 and (D5) < 1.7<br>5.9 < (D1) < 6.5 and 1.5 < (D4) < 2.4 and (D5) > 1.7<br>5.9 < (D1) < 6.5 and (D4) > 2.4<br>(D1) > 6.5 |

**Table 6 Sample of the decision tree set of rules for the gridded classification (D1, D2, D3, D4 and D5 correspond to distance to kernel**
**of class 1, 2, 3, 4 and 5). As an example, for class 1, if the distance to kernel 1 (D1) is below 3.5 and the distance to kernel 2 (D2) is above 2.2, then the grid cell belongs to class 1.**



| | | AREA | | T | P | $I_s$ | | $S_{P1.5}$ | | $P_{25\%}$ | | $S_{Tm}$ | | $I_{Arid}$ | | $T_{25\%}$ | | $S_{PET}$ | |
|---|---|---|---|---|---|---|---|---|---|---|---|---|---|---|---|---|---|---|---|
| **BASELINE 1970-2000** | CLASS 1 | 5% | | | | 0.99 | | 3.53 | | 1.70 | | 9.10 | | 39.80 | | 1.33 | | 0.00 | |
| | CLASS 2 | 18% | | | | 0.98 | | 3.88 | | 1.94 | | 8.76 | | 9.18 | | 1.32 | | 1.00 | |
| | CLASS 3 | 27% | | | | 0.87 | | 2.90 | | 1.58 | | 5.98 | | 1.75 | | 1.48 | | 4.85 | |
| | CLASS 4 | 22% | | | | 0.61 | | 0.77 | | 1.29 | | 5.81 | | 2.58 | | 1.47 | | 3.05 | |
| | CLASS 5 | 28% | | | | 0.41 | | 0.29 | | 1.20 | | 3.66 | | 0.89 | | 1.94 | | 7.56 | |
| **ALADIN RCP 4.5 2070-2100** | CLASS 1 | 4% | *0%* | 2.13 | 19% | 0.99 | *0%* | 3.45 | *-2%* | 1.79 | *6%* | 9.00 | *-1%* | 39.46 | *-1%* | 1.32 | *-1%* | 0.00 | *0%* |
| | CLASS 2 | 19% | *1%* | 2.14 | 12% | 0.98 | *0%* | 3.65 | *-6%* | 1.99 | *3%* | 8.43 | *-4%* | 9.94 | *8%* | 1.31 | *-1%* | 0.99 | *-1%* |
| | CLASS 3 | 26% | *-1%* | 2.26 | 1% | 0.87 | *0%* | 2.90 | *0%* | 1.60 | *1%* | 5.91 | *-1%* | 2.01 | *15%* | 1.44 | *-3%* | 4.51 | *-7%* |
| | CLASS 4 | 23% | *0%* | 2.14 | 2% | 0.66 | *9%* | 1.31 | *70%* | 1.32 | *3%* | 5.64 | *-3%* | 2.73 | *6%* | 1.44 | *-2%* | 2.80 | *-8%* |
| | CLASS 5 | 28% | *0%* | 2.21 | 7% | 0.45 | *7%* | 0.50 | *71%* | 1.22 | *2%* | 3.67 | *0%* | 1.06 | *20%* | 1.87 | *-4%* | 7.11 | *-6%* |
| **ALADIN RCP 8.5 2070-2100** | CLASS 1 | 4% | *0%* | 3.80 | 11% | 0.99 | *0%* | 3.44 | *-3%* | 1.83 | *8%* | 8.90 | *-2%* | 38.43 | *-3%* | 1.32 | *-1%* | 0.00 | *0%* |
| | CLASS 2 | 19% | *1%* | 3.79 | 8% | 0.98 | *0%* | 3.60 | *-7%* | 1.98 | *2%* | 8.45 | *-4%* | 10.11 | *10%* | 1.31 | *-1%* | 1.02 | *2%* |
| | CLASS 3 | 26% | *-1%* | 3.84 | -3% | 0.86 | *0%* | 2.79 | *-4%* | 1.58 | *0%* | 5.92 | *-1%* | 2.08 | *19%* | 1.44 | *-3%* | 4.38 | *-10%* |
| | CLASS 4 | 24% | *2%* | 3.76 | -2% | 0.65 | *7%* | 1.33 | *74%* | 1.34 | *4%* | 5.57 | *-4%* | 2.66 | *3%* | 1.44 | *-2%* | 2.95 | *-3%* |
| | CLASS 5 | 26% | *-2%* | 3.67 | 3% | 0.45 | *9%* | 0.57 | *96%* | 1.23 | *3%* | 3.62 | *-1%* | 0.93 | *4%* | 1.89 | *-3%* | 7.32 | *-3%* |
| **CCLM RCP 4.5 2070-2100** | CLASS 1 | 5% | *0%* | 2.67 | 6% | 0.99 | *0%* | 3.51 | *-1%* | 1.72 | *2%* | 9.06 | *0%* | 38.26 | *-4%* | 1.30 | *-2%* | 0.00 | *0%* |
| | CLASS 2 | 19% | *1%* | 2.58 | 6% | 0.99 | *0%* | 3.87 | *0%* | 2.02 | *4%* | 8.43 | *-4%* | 9.01 | *-2%* | 1.28 | *-3%* | 1.05 | *5%* |
| | CLASS 3 | 27% | *-1%* | 2.89 | -3% | 0.90 | *4%* | 3.31 | *14%* | 1.68 | *6%* | 5.97 | *0%* | 1.96 | *12%* | 1.39 | *-6%* | 4.59 | *-5%* |
| | CLASS 4 | 17% | *0%* | 3.00 | -7% | 0.77 | *27%* | 1.69 | *120%* | 1.37 | *7%* | 5.86 | *1%* | 2.89 | *12%* | 1.40 | *-5%* | 2.97 | *-3%* |
| | CLASS 5 | 32% | *0%* | 3.07 | -7% | 0.59 | *42%* | 1.22 | *316%* | 1.31 | *9%* | 3.93 | *8%* | 1.39 | *56%* | 1.69 | *-13%* | 5.70 | *-25%* |
| **CCLM RCP 8.5 2070-2100** | CLASS 1 | 5% | *0%* | 4.66 | 6% | 0.99 | *0%* | 3.57 | *1%* | 1.85 | *9%* | 8.48 | *-7%* | 36.64 | *-8%* | 1.27 | *-4%* | 0.00 | *0%* |
| | CLASS 2 | 17% | *1%* | 4.61 | 1% | 0.99 | *1%* | 3.89 | *0%* | 2.09 | *8%* | 8.10 | *-8%* | 9.49 | *3%* | 1.26 | *-4%* | 1.00 | *0%* |
| | CLASS 3 | 36% | *-1%* | 5.22 | -11% | 0.93 | *8%* | 3.40 | *17%* | 1.76 | *11%* | 5.82 | *-3%* | 2.19 | *25%* | 1.36 | *-8%* | 4.02 | *-17%* |
| | CLASS 4 | 15% | *2%* | 5.31 | -18% | 0.84 | *39%* | 2.41 | *215%* | 1.50 | *17%* | 5.60 | *-4%* | 3.20 | *24%* | 1.38 | *-6%* | 2.64 | *-13%* |
| | CLASS 5 | 28% | *-2%* | 5.36 | -23% | 0.75 | *80%* | 1.79 | *514%* | 1.39 | *16%* | 3.69 | *1%* | 1.40 | *58%* | 1.48 | *-24%* | 5.61 | *-26%* |

**Table 7 Climatic indices values under RCM ALADIN and CCLM RCP scenarios with *evolution ratio in italic***