# Peer review of "Specific Climate Classification for Mediterranean Hydrology and Future Evolution Under Med-CORDEX RCM Scenarios"

_Hydrology and Earth System Sciences, 2020_

## Referee Comment (RC1) · Christophe Cudennec (Referee) · 30 Apr 2020

The draft paper is promising as it addresses important hydroclimatic questions, with innovative approaches over the whole Mediterranean basin. Methodology and results are convincing and of high quality. The presentation and discussion can be improved.

Perspectives are strong for follow ups on the hydrological aspects as announced in the conclusion. Yet it is here essentially dealing with climatic aspects and the links and orientations to hydrology are claimed with few justification. These should be made more explicit: how are the chosen climatic indices "hydrology driven" (section 3.1)? And how are the indices dealing with precipitation seasonality specific to the Mediterranean

hydroclimatic/hydrometeorological context (L203)?

The main hydrological perspective is the catchment approach but this remains an analysis of climatic indices.

This catchment based approach is announced to be the principal objective and then the grid based approach is developed for comparison ; and monitoring stations are used for verification. The major conclusion is that there are some shifts between the catchments' mapping and the grid mapping from one class to the other. I would definitely suggest to reorganize sections 3 to present grid and station approaches first before presenting the catchment approach. Indeed the approaches with higher resolutions are the references away from which the space-averaging catchment approach shifts - not the opposite. This would further allow to have a stronger discussion of the catchment approach: averaging of indices' values over the catchment area (much simplifying vs. hydromet/hydroclim space-time dynamics in such contexts), induced clustering, effect of the catchment size, under-consideration of intra-catchment heterogeneity and eventual gradients....

One conclusion is the "continuity" of climate indices and classes, essentially North-South. This needs elaboration as continuity is not unique across the spectrum of climate classes and as the geography of climate is known to depend on both latitude, distance to the sea, and relief; and structures of gradients are season-dependent (see for instance Slimani et al., 2007; Baccour et al., 2012 and Feki et al., 2012 about geographic-seasonal structures across North-South elongated Tunisia from subhumid context to Sahara / and Vicente-Serrano et al., 2004 and follow ups in Eastern Spain)

A major question here is the boundary of the Mediterranean region. The hydrographic choice is well justified and compared to other approaches. Merheb et al. 2016 is refered to in Figure 1 but should be more strongly mobilized in section 2.1 as the antecedent study using the hydrographic definition and delineation of the region.

In that studied Mediterranean region, the climatic heterogeneity is introduced via the

Köppen classification. Yet only Köppen C classes are introduced and mapped in Figure 1 whereas part of the region falls under Köppen B classes. See the Peel et al 2007 world mapping of the Köppen classification. Figure 1 and the corresponding text should be completed to display and define all the climatic classes present in the study region. Further, the text is ambiguous at some places, suggesting Köppen identified some mediterranean climate, which does not appear in Peel et al 2007. This should be clarified.

Concerning hydroclimate classification, a further reference to the PUB body of literature could be made, such as e.g. Wagener et al., 2007; Hrachowitz et al., 2013

Details - L7: Mediterranean ... Region, basin?

- L11: convoluted sentence

- Figure 1: country boundaries are not visible. Names of countries are not all displayed (and inconsistently with other figures). See comment on mapping of other Köppen classes in the studied region.

- L114: etymology of "Mediterranean" does not mean "middle land" but "in the middle of lands"

- L13 and 123 are redundant. Some parts of Australia are also considered as "Mediterranean climate"

- Some mediterranean references about the mediterranean ground of the Turc formula should be more relevant than the Diouf et al one dealing with Senegal

- section 4.6: figure 7 and table 7 should be announced earlier in the section

- Peel et al., 2007: the final paper in HESS shall be cited, not the discussion version in HESSD

References Baccour H. et al., C, 2012. Etude synoptique conjointe des structures spatiales de l'évapotranspiration et de variables climatiques corrélées en Tunisie. Hydrological Sciences Journal, 57, 4, 818-829, DOI: 10.1080/02626667.2012.672986.

Feki H. et al., 2012. Incorporationg elevation in rainfall interpolation in Tunisia using geostatistical methods. Hydrological Sciences Journal, 57, 7, 1294-1314, DOI:10.1080/02626667.2012.710334.

Hrachowitz et al., 2013. A decade of Predictions in Ungauged Basins (PUB) – a review. Hydrological Sciences Journal, 58, 6, 1198-1255, DOI: 10.1080/02626667.2013.803183.

Merheb et al., 2016. Hydrological response characteristics of Mediterranean catchments at different time scales: a meta-analysis, Hydrological Sciences Journal, 61:14, 2520-2539, DOI: 10.1080/02626667.2016.1140174

Peel, M. C. et al., 2007. Updated world map of the Köppen-Geiger climate classification, Hydrol. Earth Syst. Sci., 11, 1633–1644, https://doi.org/10.5194/hess-11-1633-2007. Drought patterns in the Mediterranean area: the Valencia region (eastern Spain)

Vicente-Serrano, JC et al., 2004. Drought patterns in the Mediterranean area: the Valencia region (eastern Spain). Climate Research.

Slimani et al, 2007. Structure du gradient pluviométrique de la transition Méditerranée–Sahara en Tunisie: déterminants géographiques et saisonnalité. Hydrological Sciences Journal, 52, 6, 1088-1102, doi: 10.1623/hysj.52.6.1088

Wagener et al., 2007 Catchment classification and hydrologic similarity. Geography Compass 1/4: 901–931, DOI: 10.1111/j.1749-8198.2007.00039.x

---

## Referee Comment (RC2) · Anonymous Referee #2 · 11 May 2020

I am not very familiar with the climate subject. However, I find the paper very interesting. The methodology applied is very detailed, described and useful for other applications. Although the paper is well written, the take-home message is not clear. The paper claims to link climatic to hydrology. However, the link is not described. Some examples should be given to illustrate the link and the importance of hydrological modelling. I think a flowchart with the step would improve the paper readability and facilitate the application of the proposed methods. The authors need to elaborate more on the discussion in how their results affect the classification of climate and further on the predominant hydrological processes. Indeed, climate continuity seems vague. I have some questions for clarifying:

- What are the effects of the data resolution on the classification?

- How did you decide on the number of trees?

- Why you need to reduce de number of indices, "random forest" does not do that already?

- A more philosophical one: why not "R" or "Python"? The classification and machine learning methods are excellent, and it is easy to implement.
* * *

---

## Author Comment (AC1) · 17 Jun 2020

Dear Prof. Cudennec,

We are very grateful for your constructive comments on our article "Specific Climate Classification for Mediterranean Hydrology and Future Evolution Under Med-CORDEX RCM Scenarios". We totally agree with all major and specific comments and recommendations. We propose to modify the manuscript in order to respond to all points raised especially in sections 3 and 4. We will reorganize the structure of these sections to present the grid based classification as reference to the catchment based and station based classification; and we will elaborate the discussion and conclusion correspondently. All the references mentioned by the referee will be added in their corresponding section, in addition other articles will be added to justify the approach and methodology and discuss the obtained results.

In this letter (see attached supplement), the comments of Referee #1 are given in black and our responses in blue.

Kind regards,

Antoine Allam, Roger Moussa, Wajdi Najem, Claude Bocquillon

Overview

The draft paper is promising as it addresses important hydroclimatic questions, with innovative approaches over the whole Mediterranean basin. Methodology and results are convincing and of high quality. The presentation and discussion can be improved. Perspectives are strong for follow ups on the hydrological aspects as announced in the conclusion. Yet it is here essentially dealing with climatic aspects and the links and orientations to hydrology are claimed with few justifications. These should be made more explicit: how are the chosen climatic indices "hydrology driven" (section 3.1)? And how are the indices dealing with precipitation seasonality specific to the Mediterranean hydroclimatic/hydrometeorological context (L203)?

We thank Referee #1 for all his comments and agree that the choice of the climatic indices and their relation to hydrology and precipitation seasonality should be justified and detailed furthermore. The climatic, physiographic and hydrological characterisation of Mediterranenan catchments are studied and discussed in the thesis of Allam which this article constitutes the first publication. The climatic indices resulting from this classification could be the basis for a hydrological characterisation of Mediterranean catchments using multivariate analysis, useful for agriculture, ecohydrology, economy.

We suggest modifying the text in section 3.1 as follows: "The climatic indices were inspired from Köppen's definition of Mediterranean climates Csa and Csb to emphasize the precipitation and temperature variability between seasons and from the components of the water balance in its general form P=Q+E (Where P = Precipitation, Q = Runoff, E = Evapotranspiration) to highlight the link between climate and hydrology. Hence Group I and III indices (Is, P25%, P75% and IDecal) characterize Mediterranean Precipitation P in its seasonality and monthly distribution. Group II and IV indices (SPET, IArid T25% and T75%) characterize the hydrological loss to evapotranspiration in the Mediterranenan." "The precipitation seasonality characterizing the Mediterranean climate is reflected in the flow regimes of Mediterranean rivers as pointed out by Haines (1988)"

The main hydrological perspective is the catchment approach, but this remains an analysis of climatic indices. This catchment based approach is announced to be the principal objective and then the grid based approach is developed for comparison; and monitoring stations are used for verification. The major conclusion is that there are some shifts between the catchments' mapping and the grid mapping from one class to the other. I would definitely suggest to reorganize sections 3 to present grid and station approaches first before presenting the catchment approach. Indeed the approaches with higher resolutions are the references away from which the space-averaging catchment approach shifts - not the opposite. This would further allow to have a stronger discussion of the catchment approach: averaging of indices' values over the catchment area (much simplifying vs. hydromet/hydroclim space-time dynamics in such contexts), induced clustering, effect of the catchment size, under-consideration of intra-catchment heterogeneity and eventual gradients....

We agree. Similar classifications could be carried out to different hydrological or spatial units according to the needed resolution and applied field (parcels, altitudes). Sections 3 and 4 will be reorganized to show the grid based classification as reference classification, verify it for the station and catchment scale classifications.

In Section 3.6 we suggest reorganizing the framework as follows: "The proposed

methodology consisted first on calculating the grid based climatic indices using WorldClim-2 monthly data, second on reducing the number of indices with the PCA and third on classifying it using K-Means clustering. The gridded indices classification was later verified on the ground stations indices and then compared to the catchment scale averaged data for future hydrological applications. In addition, a hierarchical decision tree was constructed to avoid repeating the whole process when classifying projected indices. All PCA, K-Means and decision tree where calculated using SPSS software. Projected indices under RCP scenarios were calculated and classification evolution were then deduced."

In Section 4, we suggest reorganizing the results to show the actual Figure 5 first, then Figure 6 and last Figure 4. The sections will be reorganized in the following order Section 4.1 "PCA results for WorldClim-2 grid based indices " Section 4.2 "Grid based classification" Section 4.3 Verification on stations indices Section 4.4 Comparison to catchment based classification

One conclusion is the "continuity" of climate indices and classes, essentially North-South. This needs elaboration as continuity is not unique across the spectrum of climate classes and as the geography of climate is known to depend on both latitude, distance to the sea, and relief; and structures of gradients are season-dependent (see for instance Slimani et al., 2007; Baccour et al., 2012 and Feki et al., 2012 about geographic-seasonal structures across North-South elongated Tunisia from subhumid context to Sahara / and Vicente-Serrano et al., 2004 and follow ups in Eastern Spain)

We agree. We propose to add the suggested references and the below text to the introduction. "A North/South general precipitation and evapotranspiration gradient has been identified in Tunisia through the analysis of directional variograms that results from partial gradients evolving through seasons (Slimani et al., 2007; Baccour et al., 2012; Feki et al., 2012). These spatial gradients mainly depend on topographic structures through the interception of rainfall-generating air masses is shown. The climatic classification will try to identify the general spatial gradients across the Mediterranean."

[Figure]

A major question here is the boundary of the Mediterranean region. The hydrographic choice is well justified and compared to other approaches. Merheb et al. 2016 is referred to in Figure 1 but should be more strongly mobilized in section 2.1 as the antecedent study using the hydrographic definition and delineation of the region. Ok. We will include Merheb et al. 2016 in section 2.1 to support our choice of the hydrographic delineation of the region; with the slight difference that Merheb has considered that any catchment falling within one of the 4 definitions he advanced is considered as "Mediterranean" We proposed to modify the text in section 2.1 as follows"

"The question that arises is how would the Mediterranean boundary be defined? Several definitions of the Mediterranean boundary have been previously mentioned by Merheb et al. (2016) as collected from literature; hydrological boundary was adopted for this study as shown in Figure 1." In that studied Mediterranean region, the climatic heterogeneity is introduced via the Köppen classification. Yet only Köppen C classes are introduced and mapped in Figure 1 whereas part of the region falls under Köppen B classes. See the Peel et al 2007 world mapping of the Köppen classification. Figure 1 and the corresponding text should be completed to display and define all the climatic classes present in the study region.

Ok. Only Köppen C classes were shown to delineate the "Mediterranean climate Boundary" as Köppen B Classes are present in the region but represent the desertic climate. However, we will display all Köppen classes found in the Mediterranean in Figure 1 and complete the correspondent text with their definitions. We propose to modify the text as follows "The (Cs) climate doesn't reign all over the Mediterranean region as Köppen (B) classes are also observable. (B) classes correspond to Arid climate in general with (BWh) the Desertic and hot climate that dominates Egypt and Libya characterised with very low precipitation (MAP < 5xPth with Pth = 2xMAT) and high temperature (MAT $\geq$ 18°C), (BSk) the Arid Steppe cold climate that dominates Southeast Spain characterised with low precipitation (5xPth < MAP < 10xPth) and low temperature (MAT < 18°C), (Cf) the temperate climate without any dry season exists

in the regions of Thessaloniki and Veneto and (D) cold climate present when going further North" *MAT = Mean Annual Temperature, MAP = Mean Annual Precipitation, Pth = Pthreshold

Figure 1: Four Mediterranean region boundaries (Merheb et al. 2016); first administrative, second topographic (Milano 2013), third olive cultivation (Moreno 2014) and fourth climatic (Peel et al. 2007) Further, the text is ambiguous at some places, suggesting Köppen identified some Mediterranean climate, which does not appear in Peel et al 2007. This should be clarified.

Ok. The text will be clarified where necessary. Peel et al (2007) only reproduced "a new global map of climate using the Köppen -Geiger system based on a large global data set of long-term monthly precipitation and temperature station time series."

Concerning hydroclimate classification, a further reference to the PUB body of literature could be made, such as e.g. Wagener et al., 2007; Hrachowitz et al., 2013

Ok, we suggest adding the following text in the introduction "Through the classification of the Mediterranean catchments climatically and in a second step physiographically, we will be able to characterize their hydrological patterns and identify homogeneous regions which shall be useful for the prediction on ungauged basins (Wagener et al., 2007; Hrachowitz et al., 2013)."

Details

- L7: Mediterranean ... Region, basin?

Ok. The Mediterranean region. . .

- L11: convoluted sentence

Ok. The modified sentence will be divided into two: This classification is useful in following up hydrological (water resources management, floods, droughts, etc.), and ecohydrological applications such as Mediterranean agriculture. The olive cultivation

is the characteristic agriculture practice of the Mediterranean region.

- Figure 1: country boundaries are not visible. Names of countries are not all displayed (and inconsistently with other figures). See comment on mapping of other Köppen classes in the studied region.

Ok. Figure 1 was modified to include all Köppen classes available in Mediterranean region and make it consistent with other maps in the article.

- L114: etymology of "Mediterranean" does not mean "middle land" but "in the middle of lands"

Ok. Modified

- L73 and 123 are redundant. Some parts of Australia are also considered as "Mediterranean climate"

Ok. The redundancy will be removed from L123 and L. 73 will be modified to: "On the other hand, and at a global scale, some regions share a similar Mediterranean (Cs) climate such as California, Chile, South Africa and Australia (Figure 1)"

- Some Mediterranean references about the Mediterranean ground of the Turc formula should be more relevant than the Diouf et al one dealing with Senegal

Ok. We suggest adding the following references in Section 3.1 (Jensen & Allen, 2016) (Trajković & Kolaković, 2009) (Trajković & Stojnić, 2007)

Jensen, M. E., & Allen, R. G. (2016). Evaporation, evapotranspiration, and irrigation water requirements.

Trajković, S., & Kolaković, S. (2009). Evaluation of Reference Evapotranspiration Equations Under Humid Conditions. Water Resources Management, 23(14), 3057. doi:10.1007/s11269-009-9423-4

Trajković, S., & Stojnić, V. (2007). Effect of wind speed on accuracy of Turc method in

a humid climate. Facta universitatis-series: Architecture and Civil Engineering, 5(2), 107-113.

- section 4.6: figure 7 and table 7 should be announced earlier in the section

Ok. Figure 7 and table 7 will be announced at the beginning of section 4.6

- Peel et al., 2007: the final paper in HESS shall be cited, not the discussion version in HESSD

Ok. Reference adjusted.

References

All the references mentioned by the referee will be added in their corresponding section, in addition other articles will be added to justify the approach and methodology and discuss the obtained results.

Baccour H. et al., C, 2012. Etude synoptique conjointe des structures spatiales de l'évapotranspiration et de variables climatiques corrélées en Tunisie. Hy-drological Sciences Journal, 57, 4, 818-829, DOI: 10.1080/02626667.2012.672986.

Feki H. et al., 2012. Incorporationg elevation in rainfall interpolation in Tunisia using geostatistical methods. Hydrological Sciences Journal, 57, 7, 1294-1314, DOI:10.1080/02626667.2012.710334.

Hrachowitz et al., 2013. A decade of Predictions in Ungauged Basins (PUB) – a review. Hydrological Sciences Journal, 58, 6, 1198-1255, DOI: 10.1080/02626667.2013.803183.

Merheb et al., 2016. Hydrological response characteristics of Mediterranean catchments at different time scales: a meta-analysis, Hydrological Sciences Journal, 61:14, 2520-2539, DOI: 10.1080/02626667.2016.1140174

Peel, M. C. et al., 2007. Updated world map of the Köppen-Geiger climate classification, Hydrol. Earth Syst. Sci., 11, 1633–1644, https://doi.org/10.5194/hess-11-1633-2007. Drought patterns in the Mediterranean area: the Valencia region (eastern Spain)

Vicente-Serrano, JC et al., 2004. Drought patterns in the Mediterranean area: the Valencia region (eastern Spain). Climate Research.

Slimani et al, 2007. Structure du gradient pluviométrique de la transition Méditerranée–Sahara en Tunisie: déterminants géographiques et saisonnalité. Hydrological Sciences Journal, 52, 6, 1088-1102, doi: 10.1623/hysj.52.6.1088

Wagener et al., 2007 Catchment classification and hydrologic similarity. Geography Compass 1/4: 901–931, DOI: 10.1111/j.1749-8198.2007.00039.x

Please also note the supplement to this comment:
https://www.hydrol-earth-syst-sci-discuss.net/hess-2020-71/hess-2020-71-AC1-supplement.pdf

———————————————————

**Fig. 1.**

---

## Author Comment (AC2) · 17 Jun 2020

Dear Referee,

We are very grateful for your comments on our article "Specific Climate Classification for Mediterranean Hydrology and Future Evolution Under Med-CORDEX RCM Scenarios". We totally agree with all major and specific comments and recommendations. We propose to modify the manuscript in order to respond to all points raised especially about the link with hydrology and the discussion on the effects of the data resolution on the classification.

[Figure]

In this letter (see attached supplement), the comments of Referee #2 are given in black and our responses in blue.

Kind regards,

Antoine Allam, Roger Moussa, Wajdi Najem, Claude Bocquillon

Overview

I am not very familiar with the climate subject. However, I find the paper very interesting. The methodology applied is very detailed, described and useful for other applications. Although the paper is well written, the take-home message is not clear. The paper claims to link climatic to hydrology. However, the link is not described. Some examples should be given to illustrate the link and the importance of hydrological modelling. I think a flowchart with the step would improve the paper readability and facilitate the application of the proposed methods. The authors need to elaborate more on the discussion in how their results affect the classification of climate and further on the predominant hydrological processes. Indeed, climate continuity seems vague.

We thank Referee #2 for all his comments and agree that the link to hydrology should be justified and detailed furthermore, the steps will be clarified in the methodology to improve the paper readability for further applications, and the discussion will be elaborated to show the impact of the classification on the Mediterranean hydrology.

We suggest modifying the text in section 3.1 as follows to show the link of the climatic indices to hydrology:

"The climatic indices were inspired from Köppen's definition of Mediterranean climates Csa and Csb to emphasize the precipitation and temperature variability between seasons and from the components of the water balance in its general form P=Q+E (Where P = Precipitation, Q = Runoff, E = Evapotranspiration) to highlight the link between climate and hydrology. Hence Group I and III indices (Is, P25%, P75% and IDecal) characterize Mediterranean Precipitation P in its seasonality and monthly distribution.

[Figure]

Group II and IV indices (SPET, IArid T25% and T75%) characterize the hydrological loss to evapotranspiration in the Mediterranenan."

In section 3.6, we suggest modifying the methodology putting the gridded classification first, then the stations and last the catchment based one. The text will be as follows:

"The proposed methodology consisted first on calculating the grid based climatic indices using WorldClim-2 monthly data, second on reducing the number of indices with the PCA and third on classifying it using K-Means clustering. The gridded indices classification was later verified on the ground stations indices and then compared to the catchment scale averaged classification for future hydrological applications. In addition, a hierarchical decision tree was constructed to avoid repeating the whole process when classifying projected indices. All PCA, K-Means and decision tree where calculated using SPSS software. Projected indices under RCP scenarios were calculated and classification evolution were then deduced."

Specific comments

What are the effects of the data resolution on the classification?

We agree that data resolution impact on the classification should be clarified furthermore, we suggest adding the following text in section 5: "In this study, the climatic classification was applied and verified on three datasets of different resolutions, the grid based, the catchment based, and station based classification using the same climatic indices. The gridded data quality is limited by the spatial density and non-uniform distribution of the stations used for interpolation as they belong to different national and international networks, therefore, increasing the grid resolution can induce a loss of precision, data smoothing and increase uncertainty. Nevertheless, we can clearly notice that the grid based classification yielded the best resolution, however, despite the variability of the class boundaries between classifications, where some region shift from class to another, the overall classes setup was maintained from South to North."

How did you decide on the number of trees?

The maximum depth of the tree is set to 3 for Chi-square Automatic Interaction Detector (CHAID) method, however the level of the decision tree is automatically determined based on the classification rules to classify all the population, here gridded data. In this study the generated decision tree was of 3 levels and included 75 classification rules.

Why you need to reduce the number of indices, "random forest" does not do that already?

Thank you for this interesting comment, it let me deepen my knowledge in random forest. We agree that "random forest" can be used for classification however it only allows the hyperparameter tuning while PCA performs dimensionality reduction, reduces the number of indices and allows a separate analysis for each parameter as we did in this article. Therefore, even with random forest we shall be applying PCA before either random forest or K-Means classification. In addition, this article advances the approach over the results, nevertheless "random forests" could be applied in future articles.

A more philosophical one: why not "R" or "Python"? The classification and machine learning methods are excellent, and it is easy to implement.

Ok, we agree that "R" and "Python" could have been better tools, however the SPSS is widely used for statistical analysis with a built in functions like PCA, K-Means and Decision trees and easy to use Graphical User Interface. In addition, we tried to emphasize the new classification and approach more than the tool.

Please also note the supplement to this comment:
https://www.hydrol-earth-syst-sci-discuss.net/hess-2020-71/hess-2020-71-AC2-supplement.pdf
* * *
71, 2020.